# Hepatitis B virus prevalence and vaccine antibody titers in children HIV exposed but uninfected in Botswana

Kabo Baruti[1,2], Kayla Lentz[3], Motswedi Anderson[2], Gbolahan Ajibola[2], Bonolo B. Phinius[2], Wonderful T. Choga[2,4], Tshepiso Mbangiwa[2,4], Kathleen M. Powis [2,5,6], Theresa Sebunya[1], Jason T. Blackard[7], Shahin Lockman[2,5,8], Sikhulile Moyo[2,5], Roger Shapiro[2,5], Simani Gaseitsiwe[2,5] *

1 Department of Biological Sciences, Faculty of Science, University of Botswana, Gaborone, Botswana, 2 Botswana Harvard AIDS Institute Partnership, Gaborone, Botswana, 3 Harvard College, Cambridge, Massachusetts, United States of America, 4 Department of Pathology, Faculty of Health Sciences, University of Cape Town, Cape Town, South Africa, 5 Department of Immunology and Infectious Diseases, Harvard T. H. Chan School of Public Health, Boston, Massachusetts, United States of America, 6 Massachusetts General Hospital, Boston, Massachusetts, United States of America, 7 University of Cincinnati College of Medicine, Cincinnati, Ohio, United States of America, 8 Brigham and Women's Hospital, Boston, Massachusetts, United States of America

* sgaseitsiwe@bhp.org.bw, sgaseitsiwe@gmail.com

**Data Availability Statement:** The subset of data used for these analyses has been cleaned and extracted from the larger, complete database of the 'Mpepu study'. This dataset is ready for sharing

## Abstract

### Background

Botswana introduced the HBV vaccine at birth for all newborns in 2000. To the best of our knowledge, since the introduction of HBV vaccination, there have been limited data for vaccine response to HBV and its impact on early childhood HBV infections among children HIV exposed but uninfected in Botswana.

### Aims

To determine the prevalence of hepatitis B surface antigen (HBsAg) and HBV vaccine response in 18 months old children HIV exposed but uninfected in Botswana.

### Methods

Stored plasma samples from 304 children at 18 months of age and 287 mothers from delivery were tested for HBsAg. Mothers with positive HBsAg had HBV DNA level tested, and their HBV genotypes were determined by amplifying a 415-base pair (bp) region of the surface gene. Plasma samples from children exposed to HIV were tested for hepatitis B surface antibody (anti-HBs) titers.

### Results

No children (0 of 304) were positive for HBsAg at 18 months while 5 (1.74%) of 287 HIV-positive mothers were HBsAg positive. Four of the HBsAg positive mothers were infected with genotype A1, while 1 was infected with genotype E. The median anti-HBs titer in children

with interested researchers after going through the appropriate, and mandatory, regulatory procedures in accordance with Botswana National guidelines. Requests for data can be made by contacting Seeletso Mosweunyane (Head of Health Research Unit, Ministry of Health and Wellness, Botswana); email: wb.vog@enaynuewsoms.

**Funding:** This work was supported by the sub-Saharan African Network for TB/HIV Research Excellence (SANTHE), a DELTAS Africa Initiative (grant # DEL-15-006 to M.A, S.M, and S.G.),. The DELTAS Africa Initiative is an independent funding scheme of the African Academy of Sciences (AAS)'s Alliance for Accelerating Excellence in Science in Africa (AESA) and supported by the New Partnership for Africa's Development Planning and Coordinating Agency (NEPAD Agency) with funding from the Wellcome Trust (grant #107752/Z/ 15/Z) and the United Kingdom government.

**Competing interests:** The authors have declared that no competing interest exists.

was 174 mIU/mL [QR: 70, 457]. Three (1.1%) of 269 children had an inadequate vaccine response (<10 mIU/mL), while 266 (98.9%) of 269 had protective immunity. However, when using the $\geq$100mIU/mL threshold, only 170 (63.2%) of 269 children had complete protection.

## Conclusion

No HBsAg positivity was identified in a cohort of children HIV exposed but uninfected. The absence of HBsAg positives was associated with good HBV vaccine responses and low maternal HBsAg prevalence in Botswana.

## Introduction

Hepatitis B virus (HBV) infection is a global health problem, with 257 million people estimated to be chronically infected [1]. HBV is responsible for 887,000 deaths per year, mostly from complications such as cirrhosis and hepatocellular carcinoma [1]. In Botswana, the prevalence of HBsAg in human immunodeficiency virus (HIV)-infected individuals was 9.3% [2], while the prevalence among HIV-infected pregnant women has been reported as 3.1% [3]. HIV/ HBV co-infection is associated with high HBV viral loads and high hepatitis B envelope antigen (HBeAg) positivity with a rapid progression to cirrhosis [4].

In HBV endemic regions including sub-Saharan Africa, HBV infections may be transmitted vertically from mother to child, although most infections occur through horizontal transmission in early childhood [5]. These horizontal infections are more likely to lead to chronic infections, increasing the risk of end stage liver disease (ESLD) [5]. Globally, a meta-analysis study showed that 42.1% of the children born to HBsAg-positive mothers who did not receive HBV passive-active immunoprophylaxis acquired infection perinatally [6]. This figure was reduced to 2.9% among children who received the immunoprophylaxis, thereby highlighting the significant benefit of immunization [6]. Children become exposed to HIV in utero or via breast milk but may remain HIV-uninfected; however, there are limited data on the risk of HBV transmission via breastfeeding [7]. HBV vaccine response among children exposed to HIV has been reported to be less robust compared to children born to mothers without HIV. This places children exposed to HIV at a higher risk of HBV transmission in the presence of high viral DNA levels in HIV/HBV co-infected mothers [8].

Childhood HBV acquisition is prevented by the timely administration of the recombinant subunit vaccine to newborns [9]. Botswana adopted the World Health Organisation (WHO) recommendation in 2000 to administer a birth dose within 24 hours of birth followed by three additional doses given at 2, 3 and 4 months of life to prevent perinatal and early horizontal HBV transmission [10]. In 2015, Botswana reported a 95% national HBV vaccine coverage, but its timely administration coverage was 74% [9,11].

Infant feeding guidelines by Botswana Ministry of Health and Wellness (MoHW) in 2011 recommended that women living with HIV use formula feeding for the first 6 months of life only when it was acceptable, feasible, affordable, sustainable and safe (AFASS) [12]. In instances where formula feeding was not AFASS, mothers living with HIV were recommended to breastfeed their infants [12].

To the best of our knowledge, data on HBV vaccine responses in children HIV exposed but uninfected in Botswana remain unknown. We sought to determine the prevalence of HBsAg

positivity and vaccine response in 18 months old children HIV exposed but uninfected in Botswana.

## Methods

### Study participants'

This was a retrospective, cross-sectional study utilizing archived plasma samples from the Mpepu study conducted between 2011 and 2013 in Botswana. Residual plasma samples from the Mpepu study were stored at -80˚C in an ultra-low freezer after completion of the study. The objective of the Mpepu study was to assess the efficacy and safety of infant cotrimoxazole versus placebo prophylaxis taken from 14–18 days through 15 months of life in children exposed to HIV [13]. In this study, samples were collected at enrollment (14–28 days after birth) to 18 months of life for children, and at delivery for the mothers. The study was conducted at Botswana Harvard AIDS Institute Partnership. Mothers provided written informed consent for participation in the Mpepu study, and for storage of residual samples. Ethical approval for the study was provided by the University of Botswana ethics review committee and the Health Research Development Committee of the Botswana MoHW (protocol # 00547) for the Mpepu study and the sub-study. Archived plasma samples from children HIV exposed but uninfected and their mothers collected at 18 months and at delivery respectively were used for this study. The 18 months visit was chosen in order to measure anti-HBS titres after completion of HBV vaccine doses and to avoid quantifying passive maternal anti-HBS. Children HIV exposed but uninfected were defined as children without HIV who were born to mothers living with HIV.

### HBV surface antigen and surface antibody screening

Plasma samples were tested for HBsAg using a Murex HBsAg Version 3 Enzyme Linked Immunosorbent Assay (ELISA) kit (Murex Biotech, Dartford, UK). Initially reactive samples were confirmed by repeat duplicate testing following the manufacturer's instructions. Anti-HBs titers were determined from children's samples using a Monolisa™ Anti-HBs PLUS ELISA kit (Bio-Rad, Hercules, CA, USA) following manufacturer's instructions. The Anti-HBs lower limit of detection (LoD) was 2.00 mIU/mL, and the upper LoD was 1,000 mIU/mL. Recent studies have used either the $\geq$10mIU/mL or $\geq$100mIU/mL thresholds for protective immunity, but for our study we have used both thresholds for analysis of anti-HBs titers. For the $\geq$10mIU/mL threshold, anti-HBs levels of <10mIU/mL were considered inadequate whilst >10 mIU/mL were considered protective [14]. For the $\geq$100mIU/mL threshold, anti-HBs levels <10mIU/mL were classified as no protection, 10 mIU/mL- 100 mIU/mL anti-HBs levels were considered as partial immunity whilst anti-HBs titres of >100 mIU/mL were classified as complete protection [15].

### HBV viral load

HBV viral load was tested on HBsAg positive mothers samples collected at delivery, using COBAS® AmpliPrep/COBAS® TaqMan® HBV Test, version 2.0 (Roche, Mannheim, Germany) with a lower LoD of 20 IU/mL.

### DNA extraction

DNA was extracted from 200 μL of plasma samples using the QiAmp DNA extraction kit (QIAGEN, Hilden, Germany) according to the manufacturer's protocol with a final elution volume of 50 μL.

## HBV genotyping

A 415 base pair fragment of the HBV surface gene was amplified using a semi-nested polymerase chain reaction (PCR) as previously described [2]. First round PCR used HBV 381 primer (5′ – TGC GGC GTT TTA TCA TCT TCC T–3′; nucleotide [nt] 381–402) and HBV 840 primer (5′ – GTT TAA ATG TAT ACC CAA AGA C–3′; nt 840–861), while the second round used HBV381 and HBV801 (5′– CAG CGG CAT AAA GGG ACT CAA G–3′; nt 801–822) primers. The reaction products of the nested PCR were visualized on a 1% agarose gel stained with ethidium bromide. The PCR products were purified using the QIAquick® PCR purification kit (QIAGEN, Mannheim, Germany) according to manufacturer's specification with an elution volume of 30μL. Sequencing PCR was done using primers HBV 381 and HBV 801 and cleaned up using ZR DNA Sequencing Clean-up Kit™ (Zymo, Irvine, CA, USA). Sequencing was done using Big-dye sequencing chemistry on an ABI 3130xl sequencing machine according to the manufacturer's instructions.

## Data analysis

Sequence chromatographs were edited using Sequencher version 5.0 [16] to generate consensus sequences, and sequence alignments were created in Clustal X 2.1 [17]. Sequences aligned and manually edited using BioEdit version 6.0 [18]. Genotypes and resistance mutations were determined using the Stanford database [19]. Genotypes were confirmed using the Geno2-pheno [20], an online tool which determines the genotypes, resistance mutations and further provides sub-genotypes.

A phylogenetic tree was constructed using Bayesian Evolutionary Analysis by Sampling Trees BEAST v.1.10.4 [21] program with a chain length of 50,000,000 and sampling every 5,000 generations. The analysis utilized an uncorrelated relaxed clock, the Hasegawa, Kishino, and Yano (HKY) model, and the general time reversible model with gamma distributed rates of variation among sites and a proportion of invariable sites (GTR+G+I). Tracer v1.6 [21] was used to visualize results and confirm chain convergence. Every parameter had an effective sample size (ESS) > 200, and Tree Annotator v1.7.5 [21] was utilized to choose the maximum clade credibility tree after a 10% burn-in. Posterior probabilities > 90% were deemed statistically significant. A tree for sub-genotype A1 and genotype E sequences from this study and the respective GenBank references for the HBV surface region were constructed. The HBV surface region was used to determine the clustering of HBV strains from this study relative to other Botswana HBV sequences and few sequences from the rest of the world. HBV sequences were deposited into GenBank with accession numbers MN647947 –MN647951.

## Results

Samples from 304 children HIV exposed but uninfected were tested (154 females and 150 males). None of the children (0/304, 0%) had HBsAg detected.

Two hundred and twenty-nine (75%) of the 304 children had received a timely (within the first 24 hours) administration of the HBV birth vaccine, while 75 (25%) of 304 children had not received the vaccine timely (Table 1). The mean birth weight was 2.92 kg (95% CI, 2.87–2.97).

The residual volume of plasma in 35 (11.5%) of 304 children samples was inadequate to measure anti-HBs titers, so only 269 samples were available for testing. The median anti-HBs titer in 269 children was 174 mIU/mL [IQR: 70, 457]. Assessment of vaccine antibody response in children, using ≥10mIU/mL as the threshold for protective immunity, showed that 3 (1.1%) of 269 (95% CI, 0.2%, 3.2%) children had an inadequate immune response (<10 mIU/mL), while 266 (98.9%) of 269 (95% CI, 96.8%, 99.8%) had protective immunity. For children

Table 1. Demographics for children exposed to HIV (n = 304).

| Characteristic | n (%) |
| --- | --- |
| **Administration of HBV birth dose** | |
| Within 24 hours after birth | 229 (75.3%) |
| After 24 hours | 75 (24.7%) |
| **Feeding method** | |
| Breast feeding only | 68 (22.4%) |
| Formula feeding only | 235 (77.3%) |
| Both breast feeding and formula | 1 (0.3%) |

Abbreviations: HBV, Hepatitis B virus.

with anti-HBs titre of less than 10mIU/mL, 2 of 3 children had received a timely administration of the HBV birth dose, while 1 did not receive the HBV birth dose within the first 24 hours of birth. All children with anti-HBs titres of less than 10mIU/mL were male. However, for the ≥100 mIU/mL threshold, 170 (63.2%) of 269 children had anti-HBs titres of greater than 100 mIU/mL showing complete protection, 96 (35.7%) of 269 children had anti-HBs titers ranging between 10–100 mIU/mL showing partial immunity whilst 3 (1.1%) of 269 children had anti-HBs titers of <10mIU/mL classified as no protection (Fig 1).

We report a 1.74% (5/287) (95% CI, 0.6–4.0) prevalence of HBsAg in a cohort of mothers living with HIV with a mean age of 31.8 years (95% CI, 27.1–32.5) at delivery (Table 2). The

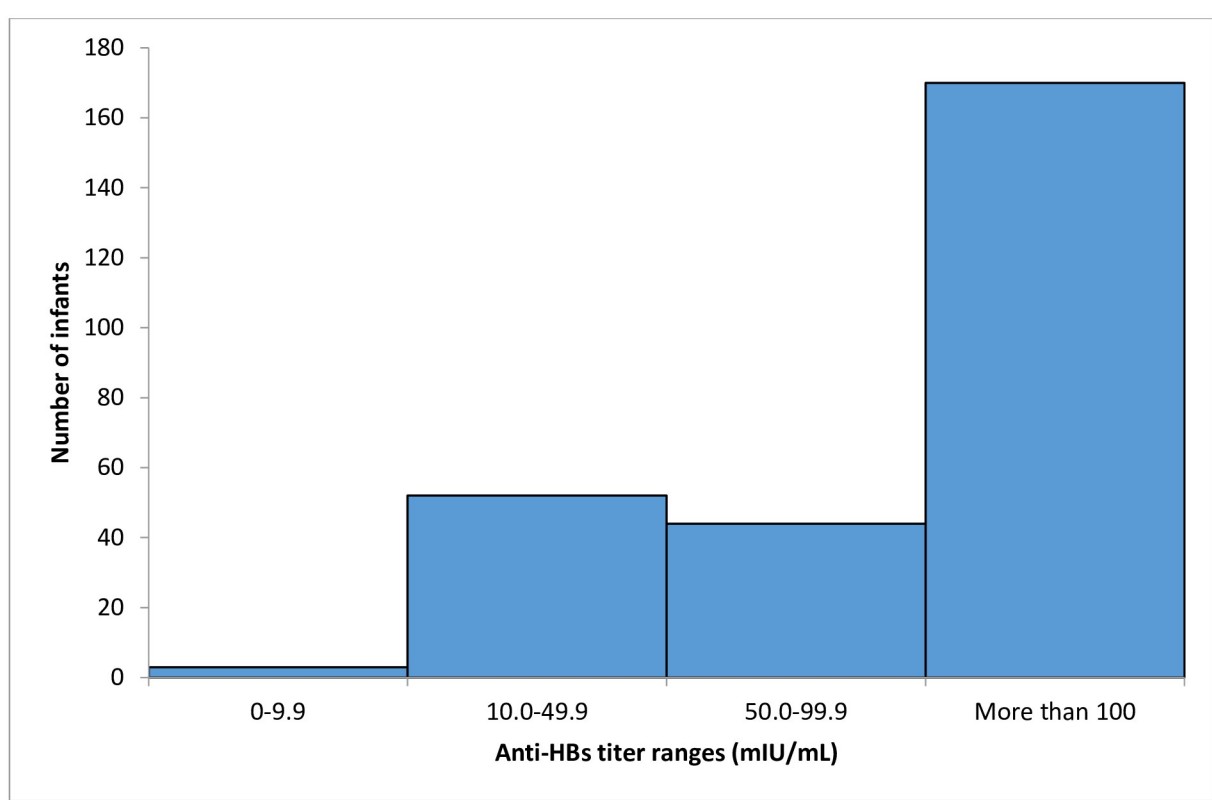

**Fig 1. Anti HBs titer ranges for children plasma samples.**

**Table 2. Demographics of mothers living with HIV.**

| | |
|---|---|
| **Mean age (years) [95% CI]** | **31.8 [31.1–32.5]** |
| **Delivery mean hemoglobulin count (cells/μL) [95% CI]** | 10.9 [10.7–11.1] |
| **Delivery median CD4+ count (cells/μL) [IQR]** | 498 [368–670] |
| **Delivery median white blood cell count (cells/μL) [IQR]** | 9.0 [7–12] |
| **Delivery median HIV viral load (IU/mL) [IQR]** | <40 [<40–691] |

Abbreviations: HIV, human immunodeficiency virus; IQR, interquartile range.

**Table 3. Viral load and ART regimen for HBsAg positive women.**

| Sample ID | Viral load | ART regimen |
|---|---|---|
| 1 | <20 IU/mL | Lopinavir/ritonavir/zidovudine/lamivudine |
| 2 | $1.13 \times 10^6$ IU/mL | Efavirenz/tenofovir/emtricitabine |
| 3 | $1.70 \times 10^8$ IU/mL | Efavirenz/tenofovir/emtricitabine |
| 4 | HBV DNA not detected | Nevirapine/emtricitabine/tenofovir |
| 5 | HBV DNA not detected | Zidovudine |

Abbreviations: HBV, hepatitis B virus; DNA, ART, antiretroviral therapy; HBsAg, hepatitis B surface antigen.

mothers had high CD4+ T-cell count (median 498 cells/μL) but low hemoglobulin and white blood cell count.

All 287 mothers living with HIV received antiretroviral treatment (ART). Table 3 shows results of HBV DNA levels for HBsAg positive mothers. The ART regimen they received included lamivudine, emtricitabine, and tenofovir which all have anti-HBV activity. Three of the 5 HBsAg positive mothers had detectable HBV DNA.

The phylogenetic tree for the five HBsAg positive mothers based on the 415bp portion of the HBsAg together with reference sequences is presented in Fig 2. Four of five sequences clustered with genotype A reference sequences, while one clustered with genotype E references.

## Discussion

This is the first study investigating HBsAg prevalence among children HIV exposed but uninfected in Botswana. We report the absence of HBsAg positivity (0%) and relatively strong HBV vaccine responses in this cohort by 18 months of age. The results in this cohort reflect public health targets promoted by the WHO African Regional Committee in 2014 where countries were challenged to reduce chronic HBV infection incidence to < 2% in children less than 5 years of age by 2020 [7].

The absence of HBsAg positivity in the children exposed to HIV cohort may be attributed to the low HBV prevalence in their mothers living with HIV as well as high HBV vaccine antibody titers in the children (Fig 1). For the ≥100 mIU/mL threshold, about 170 (63.2%) of 269 children had anti-HBs titres of more than 100 mIU/mL showing complete protection, 96 (35.7%) of 269 children had anti-HBs titers ranging between 10–100 mIU/mL showing partial immunity whilst 3 (1.1%) of 269 children had anti-HBs titers of <10mIU/mL classified as no protection. All three children with anti-HBs titers of <10mIU/mL were male which could suggest that sex-based immunological differences affected the dynamics of the immune response.

For the ≥10mIU/mL threshold, the majority of the children (98.9%) developed protective immunity, implying that the HBV vaccine given in infancy in Botswana is highly protective

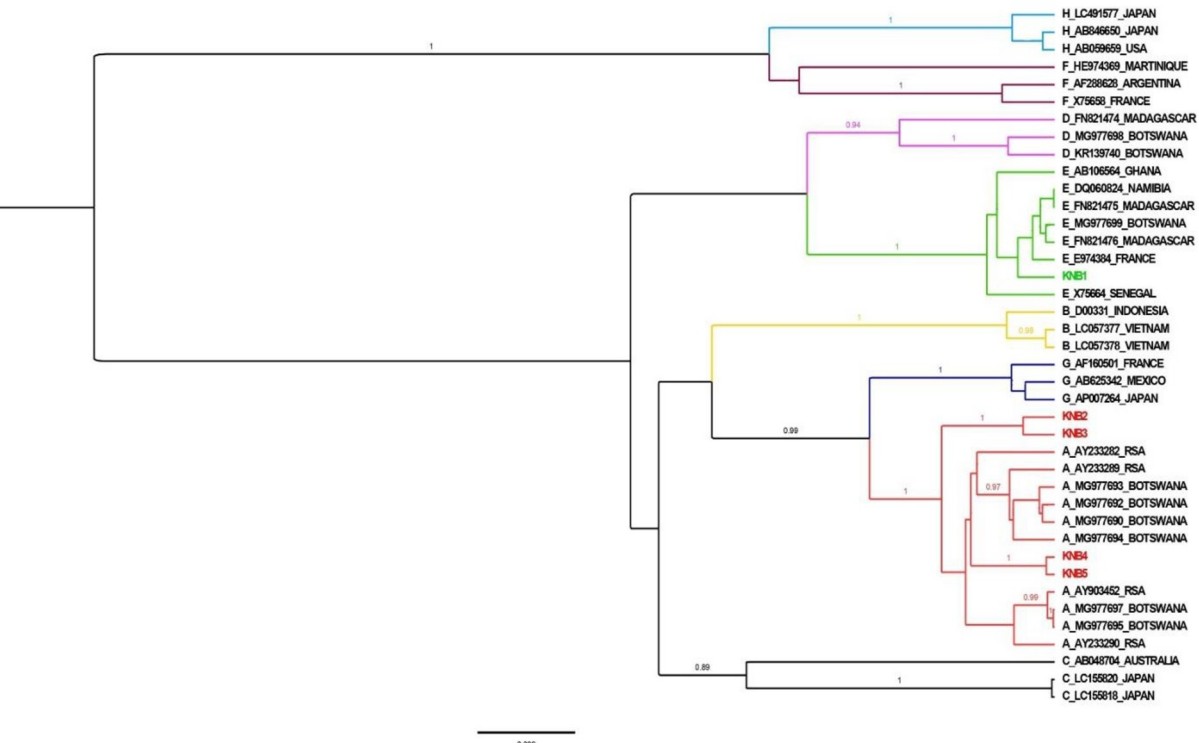

**Fig 2. Phylogenetic tree for HBsAg positive mothers sequences (highlighted in red and green) compared to reference sequences (in black).**

against chronic HBV infection. However, children with anti-HBs titres of less than 100mIU/mL may require an additional booster dose to achieve adequate immunity. A recent WHO report found that in Botswana, 94% of newborn infants complete the 3-dose schedule of HBV vaccination, while the timely hepatitis B birth dose vaccine coverage was 74% [11]. Timely administration of the hepatitis B birth dose for our study in 304 children exposed to HIV was 75% which is similar to the WHO estimates (Table 1). This HBV vaccine prevents both vertical and horizontal transmission which can occur in the first 18 months of life when children become exposed to HBV due to interaction with HBV-infected people. The median anti-HBs titer in children was 174 mIU/mL [Q1, Q3: 70, 457] which was higher than both the thresholds for protective immunity (>10 mIU/mL), and adequate protective immunity (>100mIU/mL). In Malawi, a retrospective longitudinal study was conducted to assess immune response to HBV vaccine in children exposed to HIV, who had been vaccinated at 6, 10, and 14 weeks of age and to investigate the correlation between not receiving the HBV primary vaccine dose at birth and HBV infection acquisition [8]. Results from the study reported protective (≥10mIU/mL) anti-HBs levels in 54/58 (93.2%) children at 6 months, 126/144 (87.5%) at 12 months, and 141/176 (80.1%) children at 24 months [8]. These results indicated a limited impact of HIV exposure on HBV vaccine response despite the children not receiving the HBV vaccine birth dose.

The 1.74% HBsAg prevalence in mothers living with HIV was lower than the reported prevalence from previous studies in pregnant women living with HIV which was reported as 3.1–3.8% in Botswana [3,22]. This may be because these mothers were treated with zidovudine (ZDV) and lamivudine (3TC) or tenofovir (TDF) and emtricitabine (FTC) containing ART regimens, which have anti-HBV activity (Table 3). Among the 5 mothers living with HIV who

tested positive for HBsAg, none of their children were HBsAg positive. One mother had a low HBV viral load (<20 IU/mL), while 2 had no detectable HBV DNA. The two mothers who had high HBV viral load were on efavirenz / TDF / FTC which also has dual-drug activity against HBV. However, in both mothers the ART had been initiated only a week prior to the HBV viral load sampling, which may explain the lack of apparent HBV viral load suppression. The high HBV viral load may also be attributed to acute HBV infection though this was not determined. Genotyping results showed that 4 of 5 samples from mothers living with HIV had HBV genotype A1 which is associated with high risk of horizontal transmission, and 1 sample had genotype E (Fig 2) [23]. These genotypes have been previously reported in other studies conducted in Botswana [2,3].

There were several limitations of this retrospective study. First, few children exposed to HIV were at risk for vertical HBV infection, as the HBsAg prevalence among their mothers as well as their HBV DNA levels were low. However, the risk for HBV infection at 18 months of age is not only from vertical transmissions but also from horizontal transmissions as the child has started interacting with other members of the family and playmates. Second, we did not test for occult HBV infection (i.e. negative HBsAg but positive HBV DNA infections) in children due to low volumes of the samples. This could have resulted in an underestimation of HBV infection and the long-term outcome of paediatric occult HBV infection is not well understood. Third, the HBV vaccination records of the children showed administration of the birth dose, there were no data on whether they completed all their doses which might have an effect on the HBV vaccine antibody titers.

## Conclusion

We report low HBV prevalence in mothers living with HIV and no cases of HBsAg positivity in 18 month-old children HIV exposed but uninfected in Botswana. Reassuringly, there was a high response to the HBV vaccine among children HIV exposed but uninfected in Botswana. Although a very small number of children failed to achieve protective levels of vaccine titers and might be at risk of HBV infection, these results demonstrate low risk and good vaccine protection among children exposed to HIV in Botswana, meeting the WHO-established targets for < 2% paediatric HBV prevalence in the region. However, 35.7% of the children did not achieve complete protection so further studies on this group are warranted.

## Acknowledgments

The authors would like to acknowledge Mpepu study participants.

## Author Contributions

**Conceptualization:** Kabo Baruti, Motswedi Anderson, Theresa Sebunya, Sikhulile Moyo, Roger Shapiro, Simani Gaseitsiwe.

**Data curation:** Kabo Baruti, Kayla Lentz, Motswedi Anderson, Wonderful T. Choga, Sikhulile Moyo.

**Formal analysis:** Kabo Baruti, Motswedi Anderson, Bonolo B. Phinius, Wonderful T. Choga, Tshepiso Mbangiwa, Sikhulile Moyo.

**Funding acquisition:** Shahin Lockman, Roger Shapiro, Simani Gaseitsiwe.

**Investigation:** Kayla Lentz, Motswedi Anderson, Gbolahan Ajibola, Bonolo B. Phinius, Wonderful T. Choga, Tshepiso Mbangiwa, Kathleen M. Powis, Shahin Lockman, Sikhulile Moyo, Roger Shapiro.

**Methodology:** Kayla Lentz, Motswedi Anderson.

**Project administration:** Gbolahan Ajibola, Kathleen M. Powis, Shahin Lockman, Roger Shapiro.

**Supervision:** Motswedi Anderson, Theresa Sebunya, Sikhulile Moyo, Simani Gaseitsiwe.

**Writing – original draft:** Kabo Baruti, Motswedi Anderson, Bonolo B. Phinius, Wonderful T. Choga, Tshepiso Mbangiwa, Sikhulile Moyo.

**Writing – review & editing:** Kabo Baruti, Kayla Lentz, Motswedi Anderson, Gbolahan Ajibola, Bonolo B. Phinius, Wonderful T. Choga, Tshepiso Mbangiwa, Kathleen M. Powis, Theresa Sebunya, Jason T. Blackard, Shahin Lockman, Sikhulile Moyo, Roger Shapiro.

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
