## [Decision Letter · Decision Letter 0]

14 Apr 2020

PONE-D-20-06802

Hepatitis B virus prevalence and vaccine antibody titers in HIV exposed children in Botswana

PLOS ONE

Dear Dr Gaseitsiwe,

Thank you for submitting your manuscript to PLOS ONE. After careful consideration, we feel that it has merit but does not fully meet PLOS ONE’s publication criteria as it currently stands. Therefore, we invite you to submit a revised version of the manuscript that addresses the points raised during the review process.

We would appreciate receiving your revised manuscript by May 29 2020 11:59PM. To enhance the reproducibility of your results, we recommend that if applicable you deposit your laboratory protocols in protocols.io, where a protocol can be assigned its own identifier (DOI) such that it can be cited independently in the future. For instructions see: http://journals.plos.org/plosone/s/submission-guidelines#loc-laboratory-protocols

We look forward to receiving your revised manuscript.

Kind regards,

Heather B Jaspan

Academic Editor

PLOS ONE

Journal Requirements:

2. Please provide additional details regarding participant consent. In the ethics statement in the Methods and online submission information, please ensure that you have specified what type of consent you obtained (for instance, written or verbal, and if verbal, how it was documented and witnessed).

Reviewers' comments:

Reviewer's Responses to Questions

**Comments to the Author**

1. Is the manuscript technically sound, and do the data support the conclusions?

Reviewer #1: Yes

Reviewer #2: Yes

2. Has the statistical analysis been performed appropriately and rigorously? 

Reviewer #1: Yes

Reviewer #2: Yes

3. Have the authors made all data underlying the findings in their manuscript fully available?

Reviewer #1: No

Reviewer #2: No

4. Is the manuscript presented in an intelligible fashion and written in standard English?

Reviewer #1: Yes

Reviewer #2: Yes

5. Review Comments to the Author

Reviewer #1: Summary:

This is a descriptive study reporting HBV specific antibody levels in 18-month-old HIV-exposed uninfected (HEU) infants in the Botswana setting. The manuscript shows that early immunization of infants induces antibody levels above the protective threshold in >90% of vaccine recipients, and in this cohort, no HBV infection were detected prompting good vaccine responses. The paper is well written, concise and scientifically sound and adds on existing literature regarding vaccine responses in HEU infants and thus is appropriate for this journal. Additional statistical analysis could be included to determine other factors that may be associated with infant antibody levels.

Comments

Introduction:

1. Line 69: strike out “exposure”

Methods:

1. Description of infant demographics in the study need to be included in methods or the result section.

2. Authors should describe method for classification of infants as HEU even if mentioned in the parent study.

3. Line 90: mentions sample collection at 14-28 days after birth but HB titres were not measured at this time point – need additional information why this was not done.

4. Also, was there a reason for not measuring HB titres in mothers?

5. Re-phrase line 130-133, difficult to follow.

6. Consider rephrasing line 140.

7. Line 154 “the whole surface region…surface open reading frame (ORF) was used… Do you mean surface region of HBsAg

Results:

1. No demographic description included for the children – give detail of median age at time of plasma sampling, did all infants receive all doses of the vaccine?

2. Line 164 - need a breakdown of when the infants received the vaccine – would this had correlated with antibody levels?

3. Consider re-phrasing line 165-168 for clarity.

4. Line 171 – The use of the word “about” is confusing, was 170 not the exact figure?

5. Was the range of antibody levels associated with timing of vaccination or collection of plasma samples? Are there any other maternal or infant factors associated with antibody levels e.g sex, maternal HIV viral load or CD4 levels.

6. No mention of antibody level testing in maternal samples.

Discussion:

1. Line 221: difficult to determine protection in the absence of exposure.

2. Line 226: missing the word “vaccine”

3. Line 228-230: missing reference

4. Line 240: consider rephrasing to – “..not receiving the HBV primary vaccine dose at birth”

5. Line 243 Should be in the results section.

Minor issues

1. “However” should be used to contrast preceding sentences or phrases and it’s use in Line 29, 66 and 81 is a bit misleading.

2. Double check for punctuation following in text reference inserts.

3. Line 172 – preference of the use “greater than” instead of “more than”

Reviewer #2: Intro:

Line 78: Will be helpful to mention the dosing (i.e. 6, 10 and 14week) of HBV vaccine

Also, might be important to mention the Botswana guideline as respect breast feeding among HIV infected mothers

Method:

not clear how the samples were stored (i.e. storage condition of the samples). Information might be helpful to readers

Result:

According to your analysis in line 162, two hundred and twenty nine (75%) had timely vaccination while 75 didn’t (that give a total of 304 HEU). However, in line 165 to 168, you seem to analyse the data for 269 of the 304 HEU. So what about the other 35 HEU? It may be better to have a table to visual these key results rather than have them only in text.

Why median age of mothers and not mean? Were the data not normally distributed? Could you explain this?

Despite being a retrospective study, it will be great to have some demographic information about the children: birth weight, feeding pattern (i.e. breast fed or formula fed) etc. Or at the least, reference the original published work that reported these information if available.

6. PLOS authors have the option to publish the peer review history of their article (what does this mean?). If published, this will include your full peer review and any attached files.

Reviewer #1: Yes: Sonwabile Dzanibe

Reviewer #2: Yes: OKPOKORO EVAEZI

---

## [Author Response · Author response to Decision Letter 0]

24 Jun 2020

Introduction:

1. Line 69: strike out “exposure”

“Exposure” in line 69 was removed. The sentence now reads: HIV-exposed but uninfected (HEU) children are exposed to HIV in utero or via breast milk but remain HIV-uninfected; however, there are limited data on the risk of HBV transmission via breastfeeding

Methods:

1. Description of infant demographics in the study need to be included in methods or the result section.

Table 1 under the results section has been included. The table contains infant demographics among others sex, birth weight, administration of the birth dose and mode of feeding for the infants. The Mean birth weight was also added in line 177-178.

2. Authors should describe method for classification of infants as HEU even if mentioned in the parent study.

Classification of children as HEU has been added in line 109. The sentence reads: HEU children were defined as HIV negative children who were born to HIV positive mothers.

3. Line 90: mentions sample collection at 14-28 days after birth but HB titres were not measured at this time point – need additional information why this was not done.

The reason why anti-HBs titers were not measured at this point was because we wanted to measure the titers after the infants had received all the HBV vaccine doses. In Botswana infants are vaccinated at birth, 2, and 4 months. However, we decided to test 18 months old samples to also avoid quantifying passive maternal antibodies and classifying them as vaccine antibody titers. The statement “The 18 months visit was chosen in order to measure anti-HBS titres after completion of HBV vaccine doses and to avoid quantifying passive maternal anti-HBS” was added in lines 107-108.

4. Also, was there a reason for not measuring HB titres in mothers?

Infant vaccination in Botswana started in 2000 which is 20 years ago. However, the mean age of the mothers stands at 31.8 years which means majority of them are unlikely to have received the HBV vaccine hence the decision not to test for anti-HBs titers in the maternal samples.

5. Re-phrase line 130-133, difficult to follow.

Lines 130-133, now line 141-146 were rephrased and now read: The PCR products were purified using the QIAquick® PCR purification kit (QIAGEN, Mannheim, Germany) according to manufacturer’s specification with an elution volume of 30µL. Sequencing PCR was done using primers HBV 381 and HBV 801 and cleaned up using ZR DNA Sequencing Clean-up Kit™ (Zymo, Irvine, CA, USA).

6. Consider rephrasing line 140.

Line 140, now line 152 was rephrased to: Genotypes and resistance mutations were determined using the Stanford database

7. Line 154 “the whole surface region…surface open reading frame (ORF) was used… Do you mean surface region of HBsAg

“Whole surface region” and “surface open reading frame” were used to refer to the HBV surface region. They were rephrased to HBV surface region. The sentences now in line 166-169 now read: A tree for sub-genotype A1 and genotype E sequences from this study and the respective GenBank references for the HBV surface region were constructed. The HBV surface region was used to determine the clustering of HBV strains from this study relative to other Botswana HBV sequences and few sequences from the rest of the world.

Results:

1. No demographic description included for the children – give detail of median age at time of plasma sampling, did all infants receive all doses of the vaccine?

The study only included samples collected at 18 months, which has been highlighted in line 105-107. Unfortunately, information about whether the infants received all doses was not available, we only had information about administration of the HBV birth dose which has been included in table 1 

2. Line 164 - need a breakdown of when the infants received the vaccine – would this had correlated with antibody levels?

Data about when the infants received doses given at 2, 3 and 4 months of life were not available from the parent study (Mpepu study). Only HBV birth dose information was available, provided in table 1. No correlation was found between timely administration of the HBV birth dose and antibody levels. Line 190-193 explains this: For the HEU children with anti-HBs titre of less than 10mIU/mL, 2 of 3 HEU children had received a timely administration of the HBV birth dose, while 1 did not receive the HBV birth dose within the first 24 hours of birth.

3. Consider re-phrasing line 165-168 for clarity.

Line 165-168, now line 186-190 has been rephrased and now reads: Assessment of vaccine antibody response in HEU children, using ≥10mIU/mL as the threshold for protective immunity, showed that 3 (1.1%) of 269 (95% CI, 0.2%, 3. 2%) HEU children had an inadequate immune response (<10 mIU/mL), while 266 (98.9%) of 269 (95% CI, 96.8%, 99.8%) had protective immunity.

4. Line 171 – The use of the word “about” is confusing, was 170 not the exact figure?

170 was the exact figure. The word “about” in line 194, previously line 171 has been removed. The sentence now reads: However, for the ≥100 mIU/mL threshold, 170 (63.2%) of 269 HEU children had anti-HBs titres of greater than 100 mIU/mL showing complete protection, 96 (35.7%) of 269 HEU children had anti-HBs titers ranging between 10-100 mIU/mL showing partial immunity whilst 3 (1.1%) of 269 HEU children had anti-HBs titers of <10mIU/mL classified as no protection

5. Was the range of antibody levels associated with timing of vaccination or collection of plasma samples? Are there any other maternal or infant factors associated with antibody levels e.g sex, maternal HIV viral load or CD4 levels?

The range of antibody levels was not associated with the timing of vaccination as explained in lines 190-193: For the HEU children with anti-HBs titre of less than 10mIU/mL, 2 of 3 HEU children had received a timely administration of the HBV birth dose, while 1 did not receive the HBV birth dose within the first 24 hours of birth.

The range of antibody levels could be associated with gender because all the 3 HEU children with anti-HBs titres of <10mIU/mL were male. A sentence has now been added in line 193 which reads: All the HEU children with anti-HBs titers of less than 10mIU/mL were male.

6. No mention of antibody level testing in maternal samples.

Infant vaccination in Botswana started in 2000 which is 20 years ago. However, the median age of the mothers stands at 31.8 years (table 2) which means majority of them are unlikely to have received the HBV vaccine so we decided not to test for anti-HBs titers in the maternal samples.

Discussion:

1. Line 221: difficult to determine protection in the absence of exposure.

Line 221 refers to protection as per the widely used and acceptable threshold of ≥10mIU/mL which is thought to be enough to prevent HBV transmission. The children in this study were exposed to HBV both through vertical and horizontal transmission. Vertical transmission in the sense that 5 mothers tested positive for HBsAg but did not infect their children due to their positive immune response to the vaccine. Data on the mode of feeding show that 1 HBsAg positive mother had been breastfeeding her infant, but the infant tested negative for HBsAg.

2. Line 226: missing the word “vaccine”

The word vaccine has been added to where appropriate in line 251, previously line 226. The sentence now reads: .A recent WHO report found that in Botswana, 94% of new-born infants complete the 3-dose schedule of HBV vaccination, while the timely hepatitis B birth dose vaccine coverage was 74%

3. Line 228-230: missing reference

Line 251-253, previously line 228-230, was referring to demographics of our study samples which has now been included in table 1, not from the parent Mpepu study. The line has been rephrased and now reads: Timely administration of the hepatitis B birth dose for our study in 304 HEU children was 75% which is similar to the WHO estimates (table 1).

4. Line 240: consider rephrasing to – “.not receiving the HBV primary vaccine dose at birth”

Line 261, previously line 240, has been rephrased to include “not receiving the HBV primary vaccine dose at birth”. The sentence now reads: In Malawi, a retrospective longitudinal study was conducted to assess the immune response to HBV vaccine in HIV-exposed children of unknown HIV status, who had been vaccinated at 6, 10, and 14 weeks of age and to investigate the correlation between not receiving the HBV primary vaccine dose at birth and HBV infection acquisition

5. Line 243 Should be in the results section.

Line 267, previously line 243 has been moved from the discussion and now forms part of the results (line 202-204).

Minor issues

1. “However” should be used to contrast preceding sentences or phrases and it’s use in Line 29, 66 and 81 is a bit misleading.

The word “however” has been deleted from the text. Line 30 previously line 29 reads: To the best of our knowledge, since the introduction of HBV vaccination, there have been limited data for vaccine response to HBV and its impact on early childhood HBV infections among HIV exposed uninfected (HEU) children in Botswana. Line 66 reads: This figure was reduced to 2.9% among children who received the immunoprophylaxis, thereby highlighting the significant benefit of immunization. Line 87, previously line 81 reads: To the best of our knowledge, data on HBV vaccine responses in HEU children in Botswana remain unknown.

2. Double check for punctuation following in text reference inserts.

Double checks for correct punctuation have been done and line 122-125 have been corrected.

3. Line 172 – preference of the use “greater than” instead of “more than”

“More than” has been removed from the sentence in line 195, previously line 172 and “greater than” has been added. The sentence now reads: However, for the ≥100 mIU/mL threshold, 170 (63.2%) of 269 HEU children had anti-HBs titres of greater than 100 mIU/mL showing complete protection, 96 (35.7%) of 269 HEU children had anti-HBs titers ranging between 10-100 mIU/mL showing partial immunity whilst 3 (1.1%) of 269 HEU children had anti-HBs titers of <10mIU/mL classified as no protection

Reviewer #2:

Intro:

Line 78: Will be helpful to mention the dosing (i.e. 6, 10 and 14week) of HBV vaccine

Also, might be important to mention the Botswana guideline as respect breast feeding among HIV infected mothers

Information on the dosing of the vaccine has been added. The sentence in line 76-79 now reads: Botswana adopted the World Health Organisation (WHO) recommendation in 2000 to administer a birth dose within 24 hours of birth followed by three additional doses given at 2, 3 and 4 months of life to prevent perinatal and early horizontal HBV transmission.

Data about Botswana infant feeding guidelines with respect to HIV positive mothers that were in place at the time of sampling (2011-2013) have been added in line 82-86. The sentences now read: Infant feeding guidelines by Botswana Ministry of Health and Wellness (MoHW) in 2011 recommended that HIV positive women use formula feeding for the ﬁrst 6 months of life only when it was acceptable, feasible, affordable, sustainable and safe (AFASS). In instances where formula feeding was not AFASS, HIV positive mothers were recommended to breastfeed their infants. 

Method:

not clear how the samples were stored (i.e. storage condition of the samples). Information might be helpful to readers.

Information about storage conditions of the samples have been added. The sentence in line 93-95 now reads: Residual plasma samples from the Mpepu study were stored at -80℃ in an ultra-low freezer after completion of the study.

Result:

According to your analysis in line 162, two hundred and twenty nine (75%) had timely vaccination while 75 didn’t (that give a total of 304 HEU). However, in line 165 to 168, you seem to analyse the data for 269 of the 304 HEU. So what about the other 35 HEU? It may be better to have a table to visual these key results rather than have them only in text.

During the study we encountered a problem of low volume samples for 35 of the infant samples. As a result, we did not have the required 75µl to measure anti-HBs titers in those samples. A sentence has been added in line 183-184 which reads: The residual volume of plasma in 35 (11.5%) of 304 HEU children samples was inadequate to measure anti-HBs titers, so only 269 samples were available for testing

Why median age of mothers and not mean? Were the data not normally distributed? Could you explain this?

Normal distribution has been checked in all the maternal demographics using the density histograms. Results show that only age and hgb data were normally distributed hence a change was made, in table 2, to use the mean instead of the median. Other parameters did not show normal distribution so median was used instead of the mean.

For infants, the density histogram did not show normal distribution in anti-HBs titres so the median was used. For the birth weight, the graph shows normal distribution so the mean was used instead of the median. A sentence has been added in line 177-178 which reads: The Mean birth weight was 2.92 kg (95% CI, 2.87- 2.97).

Despite being a retrospective study, it will be great to have some demographic information about the children: birth weight, feeding pattern (i.e. breast fed or formula fed) etc. Or at the least, reference the original published work that reported these information if available.

Table 1 under the results has been included. The table contains infant demographics among others sex, birth weight, administration of the birth dose and mode of feeding for the HEU infants.

---

## [Editor Report · Decision Letter 1]

29 Jun 2020

PONE-D-20-06802R1

Hepatitis B virus prevalence and vaccine antibody titers in HIV exposed children in Botswana

PLOS ONE

Dear Dr. Gaseitsiwe,

Thank you for submitting your manuscript to PLOS ONE. After careful consideration, we feel that we can provisionally accept your submission for publication. However, I request that a few additional changes are made:

Most importantly, please follow the NIAID HIV Language Guide Dated February 2020, and refer to "persons living with HIV" instead of HIV-positive as well as "Children HIV-exposed but uninfected" etc.

Additionally, Table 1 needs some editing. Also there is not need to specify both % male AND percent female unless there were some infants with unassigned sex at birth (i.e. that the % does not add up to 100).

Finally, line 177: "Mean" should not be capitalised as its in the middle of a sentence, and new-born should not be hyphenated (line 247).

We look forward to receiving your revised manuscript.

Kind regards,

Heather B Jaspan

Academic Editor

PLOS ONE

---

## [Author Response · Author response to Decision Letter 1]

9 Jul 2020

Most importantly, please follow the NIAID HIV Language Guide Dated February 2020, and refer to "persons living with HIV" instead of HIV-positive as well as "Children HIV-exposed but uninfected" etc.

The following changes have been made:

Title

Line 1-2: “HIV exposed” has been removed from the title and “children HIV exposed but uninfected” has been added. The title now reads “Hepatitis B virus prevalence and vaccine antibody titers in children HIV exposed but uninfected in Botswana”

Line 4-5: HIV positive has been removed from the short title and “children HIV exposed but uninfected” has been added. The short title now reads “HBV vaccine antibody titers in children HIV exposed but uninfected”

Abstract

Line 32: “HIV exposed but uninfected” has been removed from the sentence and “children HIV exposed but uninfected” has been added. The sentence now reads “To the best of our knowledge, since the introduction of HBV vaccination, there have been limited data for vaccine response to HBV and its impact on early childhood HBV infections among children HIV exposed but uninfected in Botswana.”

Line 35-36: “HIV exposed but uninfected” has been removed from the sentence and “children exposed to HIV” has been added. The sentence now reads “To determine the prevalence of hepatitis B surface antigen (HBsAg) and HBV vaccine response in 18 months old children HIV exposed but uninfected in Botswana.”

Line 37: “HEU” has been removed from the sentence. The sentence now reads “Stored plasma samples from 304 children at 18 months of age and 287 mothers from delivery were tested for HBsAg.”

Line 40-41: “HEU children” has been removed from the sentence and “children exposed to HIV” has been added. The sentence now reads, “Plasma samples from children exposed to HIV were tested for hepatitis B surface antibody (anti-HBs) titers.”

Line 42: “HEU” has been removed from the sentence. The sentence now reads, “No children (0 of 304) were positive for HBsAg at 18 months while 5 (1.74%) of 287 HIV-positive mothers were HBsAg positive.”

Line 46: “HEU” has been removed from the sentence. The sentence now reads, “Three (1.1%) of 269 children had an inadequate vaccine response (<10 mIU/mL), while 266 (98.9%) of 269 had protective immunity.”

Line 48: “HEU” has been removed from the sentence. The sentence now reads, “However, when using the ≥100mIU/mL threshold, only 170 (63.2%) of 269 children had complete protection.”

Line 49-50: “HEU children” has been removed from the sentence and “children HIV exposed but uninfected” has been added. “Were” has been changed to “was”. The sentence now reads,” No HBsAg positivity was identified in a cohort of children HIV exposed but uninfected.”

Introduction

Line 69-70: “HEU children” has been removed from the sentence. The sentence now reads, “Some children exposed to HIV in utero or via breast milk may remain HIV-uninfected; however, there are limited data on the risk of HBV transmission via breastfeeding.”

Line 71-73: “HEU” has been removed from the sentence and “exposed to HIV” has been added. “HIV negative” has been removed and replaced by “mothers without HIV”. The sentence now reads, “HBV vaccine response among children exposed to HIV has been reported to be less robust compared to children born to mothers without HIV.

Line 73-74: “HEU” has been removed from the sentence and “exposed to HIV” has been added. The sentence now reads, “This places children exposed to HIV at a higher risk of HBV transmission in the presence of high viral DNA levels in HIV/HBV co-infected mothers.”

Line 84: “HIV positive women” has been removed from the sentence and replaced with “women living with HIV”. The sentence now reads, “Infant feeding guidelines by Botswana Ministry of Health and Wellness (MoHW) in 2011 recommended that women living with HIV use formula feeding for the ﬁrst 6 months of life only when it was acceptable, feasible, affordable, sustainable and safe (AFASS).”

Line 86-87: “HIV positive” has been removed from the sentence and replaced with “mothers living with HIV”. The sentence now reads, “In instances where formula feeding was not AFASS, mothers living with HIV were recommended to breastfeed their infants.”

Line 88-89: “HIV exposed but uninfected” has been removed from the sentence and “children HIV exposed but uninfected” has been added. The sentence now reads “To the best of our knowledge, data on HBV vaccine responses in children HIV exposed but uninfected in Botswana remain unknown.”

Line 90-91: “HEU” has been removed from the sentence and “children HIV exposed but uninfected” has been added. The sentence now reads, “We sought to determine the prevalence of HBsAg positivity and vaccine response in 18 months old children HIV exposed but uninfected in Botswana.”

Methods

Line 99: “HEU” has been removed from the sentence and “exposed to HIV” has been added. The sentence now reads, “The objective of the Mpepu study was to assess the efficacy and safety of infant cotrimoxazole versus placebo prophylaxis taken from 14-18 days through 15 months of life in children exposed to HIV”

Line 106-107: “HEU” has been removed from the sentence and “children HIV exposed but uninfected” has been added. The sentence now reads, “Archived plasma samples from children HIV exposed but uninfected and their mothers collected at 18 months and at delivery respectively were used for this study.”

Line 110-112: “HEU” has been removed from the sentence and “children HIV exposed but uninfected” has been added. “HIV negative” has been removed from the sentence and replaced with “children without HIV”. “HIV positive” has also been removed from the sentence and replaced with “mothers living with HIV”. The sentence now reads, “Children HIV exposed but uninfected were defined as children without HIV who were born to mothers living with HIV.”

Results

Line 173: “HEU” has been removed from the sentence and “children HIV exposed but uninfected” has been added. The sentence now reads, “Samples from 304 children HIV exposed but uninfected were tested (154 females and 150 males).”

Line 175-177: “HEU” has been removed from the sentence. The sentence now reads, “Two hundred and twenty-nine (75%) of the 304 children had received a timely (within the first 24 hours) administration of the HBV birth vaccine, while 75 (25%) of 304 children had not received the vaccine timely (Table1).”

Line 179: “HEU infant” has been removed from table 1 title and “for children exposed to HIV” has been added. The title for table 1 now reads, “Demographics for children exposed to HIV (n=304)”

Line 183: “HEU” has been removed from the sentence. The sentence now reads, “The residual volume of plasma in 35 (11.5%) of 304 children samples was inadequate to measure anti-HBs titers, so only 269 samples were available for testing.”

Line 185: “HEU” has been removed from the sentence. The sentence now reads, “The median anti-HBs titer in 269 children was 174 mIU/mL [IQR: 70, 457]”

Line 186-189: “HEU” has been removed from the sentence. The sentence now reads, “Assessment of vaccine antibody response in children, using ≥10mIU/mL as the threshold for protective immunity, showed that 3 (1.1%) of 269 (95% CI, 0.2%, 3.2%) children had an inadequate immune response (<10 mIU/mL), while 266 (98.9%) of 269 (95% CI, 96.8%, 99.8%) had protective immunity”

Line 189-192: “HEU” has been removed from the sentence. The sentence now reads, “For children with anti-HBs titre of less than 10mIU/mL, 2 of 3 children had received a timely administration of the HBV birth dose, while 1 did not receive the HBV birth dose within the first 24 hours of birth”

Line 191-192: “HEU” has been removed from the sentence. The sentence now reads, “All children with anti-HBs titres of less than 10mIU/mL were male.”

Line 192-197: “HEU” has been removed from the sentence. The sentence now reads, “. However, for the ≥100 mIU/mL threshold, 170 (63.2%) of 269 children had anti-HBs titres of greater than 100 mIU/mL showing complete protection, 96 (35.7%) of 269 children had anti-HBs titers ranging between 10-100 mIU/mL showing partial immunity whilst 3 (1.1%) of 269 children had anti-HBs titers of <10mIU/mL classified as no protection (Fig 1).”

Line 199: “HEU” has been removed from the figure 1 title. The title now reads, “Fig 1. Anti HBs titer ranges for children plasma samples”

Line 202: “HIV positive” has been removed from the sentence and replaced with “mothers living with HIV”. The sentence now reads, “We report a 1.74% (5/287) (95% CI, 0.6- 4.0) prevalence of HBsAg in a cohort of mothers living with HIV with a mean age of 31.8 years (95% CI, 27.1- 32.5) at delivery (Table 2).”

Line 209: “HIV positive” has been removed from the title of table 2 and replaced with “mothers living with HIV”. The title now reads, “Table 2. Demographics of mothers living with HIV"

Line 211: “HIV positive” has been removed from the sentence and replaced with “mothers living with HIV”. The sentence now reads, “All 287 mothers living with HIV received antiretroviral treatment (ART).”

Discussion

Line 227-228: “HEU” has been removed from the sentence and “children HIV exposed but uninfected” has been added. The sentence now reads, “This is the first study investigating HBsAg prevalence among children HIV exposed but uninfected in Botswana.”

Line 228-230: “of HEU children” and “at 18 months” have been removed from the sentence. The sentence now reads, “We report the absence of HBsAg positivity (0%) and relatively strong HBV vaccine responses in this cohort by 18 months of age.

Line 231: “are” has been removed and replaced with “were” to correct the tense. The sentence now reads, “The results in this cohort reflect public health targets promoted by the WHO African Regional Committee in 2014 where countries were challenged to reduce chronic HBV infection incidence to < 2% in children less than 5 years of age by 2020.”

Line 234-236: Line 110-111: “HEU” has been removed from the sentence and “exposed to HIV” has been added. “HIV positive” has also been removed from the sentence and replaced with “mothers living with HIV”. The sentence now reads, “The absence of HBsAg positivity in the children exposed to HIV cohort may be attributed to the low HBV prevalence in their mothers living with HIV as well as high HBV vaccine antibody titers in the children (Fig 1)”

Line 236-240: “HEU” has been removed from the sentence. The sentence now reads, “For the ≥100 mIU/mL threshold, 170 (63.2%) of 269 children had anti-HBs titres of greater than 100 mIU/mL showing complete protection, 96 (35.7%) of 269 children had anti-HBs titers ranging between 10-100 mIU/mL showing partial immunity whilst 3 (1.1%) of 269 children had anti-HBs titers of <10mIU/mL classified as no protection.”

Line 241: “HEU infants” has been removed from the sentence and replaced with “children”. The sentence now reads, “All three children with anti-HBs titers of <10mIU/mL were male which could suggest that sex based immunological differences affected the dynamics of the immune response.”

Line 244: “HEU” has been removed from the sentence and replaced with “the”. The sentence now reads, “For the ≥10mIU/mL threshold, the majority of the children (98.9%) developed protective immunity, implying that the HBV vaccine given in infancy in Botswana is highly protective against chronic HBV infection.”

Line 246: “the HEU” has been removed from the sentence. The sentence now reads, “However, children with anti-HBs titres of less than 100mIU/mL may require an additional booster dose to achieve adequate immunity.”

Line 251: “HEU” has been removed from the sentence and “exposed to HIV” has been added. The sentence now reads, “Timely administration of the hepatitis B birth dose for our study in 304 children exposed to HIV was 75% which is similar to the WHO estimates (Table 1).”

Line 258-259: “HEU” has been removed from the sentence and “exposed to HIV” has been added. “Of unknown HIV status” has also been removed from the sentence, The sentence now reads, “In Malawi, a retrospective longitudinal study was conducted to assess immune response to HBV vaccine in children exposed to HIV, who had been vaccinated at 6, 10, and 14 weeks of age and to investigate the correlation between not receiving the HBV primary vaccine dose at birth and HBV infection acquisition.”

Line 266-268: “HIV positive” has been removed from the sentence and replaced with “mothers living with HIV”. The sentence now reads, “The 1.74% HBsAg prevalence in mothers living with HIV was lower than the reported prevalence from previous studies in pregnant women living with HIV which was reported as 3.1-3.8% in Botswana”

Line 271: “HIV positive” has been removed from the sentence and replaced with “mothers living with HIV”. The sentence now reads, “Among the 5 mothers living with HIV who tested positive for HBsAg, none of their children were HBsAg positive.”

Line 278-279: “from mothers living with HIV” and “HBV” have been added to the sentence which now reads, “Genotyping results showed that 4 of 5 samples from mothers living with HIV had HBV genotype A1 which is associated with high risk of horizontal transmission, and 1 sample had genotype E”

Line 282-283: “HEU” has been removed from the sentence and “exposed to HIV” has been added. The sentence now reads, “First, few children exposed to HIV were at risk for vertical HBV infection, as the HBsAg prevalence among their mothers as well as their HBV DNA levels were low.”

Line 291: “HEU” has been removed from the sentence. The sentence now reads, “Third, the HBV vaccination records of the children showed administration of the birth dose, there were no data on whether they completed all their doses which might have an effect on the HBV vaccine antibody titers.”

Conclusion

Line 296-298: “HEU” has been removed from the sentence and “exposed to HIV” has been added. “HIV positive” has also been removed from the sentence and replaced with “mothers living with HIV”. The sentence now reads, “We report low HBV prevalence in mothers living with HIV and no cases of HBsAg positivity in 18 month-old children exposed to HIV in Botswana.”

Line 298-299: “the HEU” and “in this cohort” have been removed while “exposed to HIV” has been added. The sentence now reads, “Reassuringly, there was a high response to the HBV vaccine among children exposed to HIV in Botswana.”

Line 302: “HEU” has been removed from the sentence and “exposed to HIV” has been added. The sentence now reads, “Although a very small number of children failed to achieve protective levels of vaccine titers and might be at risk of HBV infection, these results demonstrate low risk and good vaccine protection among children exposed to HIV in Botswana, meeting the WHO-established targets for < 2% paediatric HBV prevalence in the region.”

Line 304: “HEU” has been removed from the sentence. The sentence now reads,” However, 35.7% of the children did not achieve complete protection so further studies on this group are warranted.”

Additionally, Table 1 needs some editing. Also there is no need to specify both % male AND percent female unless there were some infants with unassigned sex at birth (i.e. that the % does not add up to 100).

Table 1 has been edited. Information about female percentages has been removed from table 1 because there were no infants with unassigned sex. 

Finally, line 177: "Mean" should not be capitalised as its in the middle of a sentence, and new-born should not be hyphenated (line 247).

“Mean” in line 177 has been changed to “mean”. This sentence now reads “The mean birth weight was 2.92 kg (95% CI, 2.87- 2.97).”

Hyphenation has been removed from the word “new-born” in line 248. The sentence now reads, “A recent WHO report found that in Botswana, 94% of newborn infants complete the 3-dose schedule of HBV vaccination, while the timely hepatitis B birth dose vaccine coverage was 74%.”

---

## [Editor Report · Decision Letter 2]

23 Jul 2020

Hepatitis B virus prevalence and vaccine antibody titers in children HIV exposed but uninfected in Botswana

PONE-D-20-06802R2

Dear Dr. Gaseitsiwe,

We’re pleased to inform you that your manuscript has been judged scientifically suitable for publication and will be formally accepted for publication once it meets all outstanding technical requirements.

Kind regards,

Heather B Jaspan

Academic Editor

PLOS ONE
---

## [Editor Report · Acceptance letter]

29 Jul 2020

PONE-D-20-06802R2 

Hepatitis B virus prevalence and vaccine antibody titers in children HIV exposed but uninfected in Botswana 

Dear Dr. Gaseitsiwe:

I'm pleased to inform you that your manuscript has been deemed suitable for publication in PLOS ONE. Congratulations! Your manuscript is now with our production department. 

Kind regards, 

on behalf of

Dr. Heather B Jaspan 

Academic Editor

PLOS ONE